# Heavy Load Carrying and Symptoms of Pelvic Organ Prolapse among Women in Tanzania and Nepal: An Exploratory Study

**DOI:** 10.3390/ijerph18031279

**Published:** 2021-01-31

**Authors:** Aybüke Koyuncu, Jillian L. Kadota, Agatha Mnyippembe, Prosper F. Njau, Tula Ram Sijali, Sandra I. McCoy, Michael N. Bates, Carisa Harris-Adamson, Ndola Prata

**Affiliations:** 1Division of Epidemiology & Biostatistics, University of California, Berkeley, CA 94720-7360, USA; jill.kadota@berkeley.edu (J.L.K.); smccoy@berkeley.edu (S.I.M.); m_bates@berkeley.edu (M.N.B.); 2Health for a Prosperous Nation, Dar es Salaam 0701, Tanzania; amnyippembe@gmail.com; 3Prevention of Mother-to-Child HIV Transmission Programme, Ministry of Health, Community Development, Gender, Elderly, and Children, Dar es Salaam 0701, Tanzania; prosperpendo@gmail.com; 4Institute of Social and Environment Research-Nepal (ISER-N), Kaski 33700, Nepal; sijali.tula@gmail.com; 5Division of Environmental Health Sciences, University of California, Berkeley, CA 94720-7360, USA; carisaharris@berkeley.edu; 6Department of Medicine, University of California, San Francisco, CA 94143, USA; 7Bixby Center for Population, Health and Sustainability, School of Public Health, University of California, Berkeley, CA 94720-7360, USA; ndola@berkeley.edu

**Keywords:** pelvic organ prolapse (POP), uterine prolapse, heavy load carrying

## Abstract

Heavy load carrying of water, firewood, and sand/stones is a ubiquitous activity for women living in developing countries. Although the intra-abdominal pressure associated with heavy load carrying is hypothesized to increase the risk of pelvic organ prolapse (POP) among women, relevant epidemiologic data are lacking. We conducted a comparative study involving two exploratory cross-sectional studies among convenience samples of women carrying heavy loads, with different characteristics: (1) as part of their activities for daily living, in Shinyanga region, Tanzania; and (2) working as sand miners in Pokhara, Nepal. Women were categorized has having “low” or “high” load-carrying exposures based on the measured weights of the loads being carried at the time of the survey, as well as on self-reported duration and frequency of load carrying. A summary score for lower abdominal discomfort suggestive of POP was generated using questions from the Pelvic Organ Prolapse Distress Inventory (POPDI-6). Women with higher load carrying exposures had on average higher discomfort scores in both Tanzania (adjusted prevalence difference (PD_a_) = 3.7; 95% CI: −3.8–11.3; *p* = 0.33) and Nepal (PD_a_ = 9.3; 95% CI: −4.9–23.6; *p* = 0.18). We identified trends suggestive of an association between increasing heavy load carrying exposures and symptoms of lower abdominal discomfort. Our findings underscore the need for larger epidemiologic studies of the potential adverse reproductive health effects of heavy load carrying activities on women in developing countries.

## 1. Introduction

Pelvic organ prolapse (POP) is a common reproductive health condition estimated to affect approximately 19% of women in low- and middle-income countries (LMIC) [1]. POP is caused by the weakening of muscles and tissues supporting the pelvis, ultimately resulting in one or more of the pelvic organs (e.g., uterus, bladder, rectum) dropping from their normal position [2]. Women living with POP experience chronic symptoms, such as incontinence and severe pain when walking, urinating, defecating, completing daily tasks, and during sexual activity [3]. While, globally, POP typically occurs in women of post-menopausal age, the highest prevalence of POP in LMICs is found in women of reproductive age [4]. The stigma and shame associated with POP, combined with patriarchal societal structures in many LMICs, likely contribute to an underestimation of the prevalence of POP [4,5,6]. In developing countries. such as Tanzania and Nepal, the pain and discomfort of POP are compounded by negative emotional effects from societal discrimination, as well as spousal and intrafamilial abuse due to women’s impaired abilities to complete household chores (e.g., collecting water, cooking) [4,5,6,7]. POP also increases women’s vulnerability to gender-based domestic violence if they refuse sex with their spouses, because of pain during intercourse [4,5,6]. Nonetheless, there has been little research on the extent of the problem, risk factors, or interventions.

In addition to known risk factors for POP, such as early childbearing, inadequate pregnancy spacing, and prolonged or difficult labor, heavy-load carrying has been implicated as a potential contributor to high rates of POP in developing countries [1,8,9,10,11]. Heavy lifting increases abdominal pressure and strain on the muscles and ligaments supporting pelvic organs [12,13]. In LMICs, the extent to which individuals carry heavy loads for long distances is determined by their access to water, fuel, and transportation, which is highly influenced by source of income and socio-economic status. Often considered a low-status activity in areas of Sub-Saharan Africa and Asia, domestic load carrying is predominantly performed by females starting as early as 10 years of age [14,15].

Although the magnitude, frequency and duration of the loads carried regularly by women in developing countries suggest such practices may contribute to adverse reproductive health outcomes, including POP, epidemiologic data are limited. In clinic-based descriptive studies heavy load carrying has been documented as a prevalent practice among women with POP [7,16], however few studies have examined heavy load carrying as a predictor of POP. According to mainly anthropologic research, women experiencing POP in developing countries frequently report heavy load carrying and strenuous physical labor during the postpartum period as perceived causes of POP [5,7]. In the few relevant studies conducted to date, load carrying has been identified as a risk factor for POP in some settings [17,18], but not in others [19]. A sample of women in Tanzania carrying heavy objects for ≥2 h per day had almost 5 times the odds of POP compared to those who did not carry heavy objects (95% CI: 1.7–13.2) [18], while carrying heavy objects for at least 12 h a week was not a risk factor for POP among a sample of women in Nepal [19]. This may reflect varied characteristics of load carrying by women including the weight of the load, the load carrying duration (determined by distances travelled), and the frequencies of load carrying. The type of load (wood, bricks, water) and the mode of carriage (atop head versus on back) could also contribute to the overall biomechanical strain on the musculoskeletal tissues that support the pelvic organs. Variations in mode of carriage may lead to different health-related outcomes.

To increase understanding of the association between heavy load-carrying practices and POP, we conducted two exploratory cross-sectional studies in separate populations of women with different heavy load carrying practices in Tanzania and Nepal. These countries were chosen because members of our group had existing research sites in both and because of the notably different methods of load carrying employed in those two countries.

## 2. Materials and Methods

Study populations were selected because of their different load-carrying characteristics (mode of carriage, load weights, frequency and duration of carriage). The study population in Shinyanga Region, Tanzania included women carrying loads as part of their activities for daily living (e.g., collecting water and other household necessities). In Nepal, the study population was comprised of sand miners working in Kaski district. Details about the population at the sand mining site from which our participants were recruited has been previously described [20].

In both study settings, eligible participants were a convenience sample of women, 18 years or older, passing a study recruitment site carrying a load. In Tanzania, study recruitment sites were established at locations near water collection points and along farm and market routes [21]. In Nepal, a sand mining site along the Seti River in Pokhara city served as the primary recruitment site [20].

Data collection took place from July–August 2016 and December-January 2017 in Tanzania and Nepal, respectively. Women who provided written informed consent by signature or thumbprint completed an interviewer-administered survey which collected information on socio-demographics, load-carrying practices, and reproductive health. The height and weight of each participant, as well as the weight of the load being carried at time of study recruitment, were measured upon survey completion. Interviewer observations covered the type of load being carried at the time of study recruitment and the method of load carrying (e.g., atop head, on back, etc.).

Participants were asked to report their average load-carrying duration (number of minutes per load carried) and frequency of carrying (number of loads carried per week). Data on duration, frequency, and the weight of the load carried at the time of study recruitment were used to generate a summary index of load-carrying practices using principal component analysis (PCA) [22]. Data patterns within the various simultaneously occurring dimensions of load carrying were captured using the first component of the PCA [22]. For participants in Tanzania, the summary index for load carrying incorporated the self-reported duration and frequency for all loads carried as part of daily activities (e.g., carrying loads of wood, water or other household necessities). In contrast, the summary index for the Nepali participants was constructed using self-reported duration and frequency of occupational loads of sand or stones. Separately, for both datasets, the summary index of load-carrying exposures was dichotomized using the median index score to indicate “low” or “high” load-carrying exposure index.

Additional data on load-carrying activities were collected from a sub-sample of study participants in Tanzania to corroborate self-reported load-carrying practices. Specifically, a wristwatch-like device enabled with GPS capabilities was utilized to assess distance traveled by women over a 24-h wearing period. In order to capture the number of loads carried during each 24-h period, participants were asked to press a button on the wristwatch-like device each time a new load was carried during the wearing period. Monitoring of distance traveled over a 24-h period was not conducted among study participants in Nepal, mainly because the distances traveled were much shorter (although more frequent) and within a worksite.

Symptoms of lower abdominal discomfort suggestive of POP were measured using questions from the Pelvic Organ Prolapse Distress Inventory (POPDI-6) validated scale (see Appendix A) [23]. The POPDI-6 scale is a subset of questions within the psychometrically validated Pelvic Floor Distress Inventory (PFDI-20), a condition-specific questionnaire used to measure the extent to which POP symptoms affect an individual’s quality of life among women with symptomatic POP [23,24,25]. Questions from the validated short-form version of the POPDI-6 scale, used as a screening tool in each study population, included questions on the presence of: (1) pressure in the lower abdomen, (2) heaviness/dullness in the pelvic area, and (3) the presence of a bulge in the vaginal area. Participants indicated the presence or absence of each symptom. The participant also rated the degree to which each symptom bothered her, with “0” indicating symptom not present; “1”, symptom present but not bothersome; “2”, symptom present and somewhat bothersome; “3”, symptom present and moderately bothersome; and “4”, symptom present and quite bothersome. Symptom ratings were used to generate a continuous summary score of lower abdominal discomfort suggestive of POP. Separately, in each study population, this summary score was dichotomized at the median into “low” and “high” scores. In addition to symptoms of POP captured in the POPDI-6 scale, participants were asked about any problems they had in holding urine and about pain during urination. Although urinary incontinence/pain during urination are frequent symptoms of POP [2], they were not included in the summary score of discomfort due to their low sensitivity for POP and possible association with other unrelated health conditions, such as urinary tract infections.

To gain a better understanding of the symptoms contributing most significantly to lower abdominal discomfort suggestive of POP in each study population, we examined the association between individual symptoms of lower abdominal discomfort and the overall POPDI score using linear regression models. All models were adjusted for potential confounders specified a priori as epidemiologically relevant and included continuous variables for body mass index (BMI), age, and parity.

The relationship between load-carrying exposures and lower abdominal discomfort suggestive of POP was modeled separately in each study population. Linear probability models were used to examine the unadjusted and adjusted associations between the summary indexes of load- carrying exposures and summary scores of lower abdominal discomfort suggestive of POP. Fully adjusted models included all covariates specified *a priori* (i.e., BMI, age, and parity). Using adjusted logistic regression, we examined the predicted probability of high/low lower abdominal discomfort by load-carrying exposures.

Statistical significance of parameter estimates was evaluated at the two-sided alpha level of *p* < 0.05. All analyses were conducted using STATA 14 (College Station, TX, USA).

## 3. Results

The combined study population was 102 women, including 80 in Tanzania (mean age 31.6 (SD: 12.2)) and 22 in Nepal (mean age 36.1 (SD: 8.1)). Women reported having given birth an average of 4.0 (SD: 2.5) and 3.4 (SD: 2.2) times, in Tanzania and Nepal, respectively (Table 1). Most participants had normal or overweight body mass index (BMI), and the proportion of participants who were overweight was higher in Nepal than in Tanzania (46% vs. 34%).

### 3.1. Load-Carrying Exposures

The two study populations were distinguished particularly by modes of load carriage, the sizes of their loads, and distances and frequencies of load carriage. Women from the study population in Tanzania carried loads balanced atop their heads (Figure 1), while women in Nepal carried loads using a cone-shaped basket called a “doko”, supported on their back with a circular band called a “namlo,” looped around the forehead (Figure 2). The Tanzanian population carried lighter and fewer loads, but for longer distances, while the Nepali population carried very heavy loads relatively short distances and more frequently.

Load-carrying exposures for our participants in Tanzania have been previously reported [21]. Briefly, on the interview date, women were carrying loads of water (n = 51; 62%), wood (n = 17; 21%) or agricultural products (n = 14; 17%). The average weight of all loads measured in Tanzania was 20.0 kg (SE: 0.6): water weighed an average of 20.4 kg (SE: 0.6), wood 21.0 kg (SE: 2.3), and agricultural products 15.9 kg (SE: 1.2). Women reported carrying an average of 3 loads per day in the last 7 days (SE: 0.2), corroborated by information collected using GPS devices from a sub-sample of study participants (n = 14). GPS data revealed an average of 3.6 load-carrying trips per day (range: 1–9 trips), covering an average total distance of 5.32 miles over the course of the 24-h wearing period [21]. Those categorized in the “high” load-carrying exposure index group carried a slightly heavier average load weight, but at a greater frequency compared to those in the “low” index group (weight: 20.0 kg (SE: 0.9) versus 19.2 kg (SE: 0.9); frequency: 33.5 loads/week (SE: 2.3) versus 11 loads/week (SE: 0.9), respectively).

Women in Nepal were carrying loads exclusively of sand and stones. Compared to women in Tanzania, women in Nepal carried substantially heavier loads for shorter durations and higher frequencies per week (Table 2). The median loads carried by female sand miners were 65.5 kg (SE: 2.2), and women reported carrying loads a median of 165.5 times per week (SE: 115.9) for less than 1 min per load (SE: 0). Sand-miners in the “high” heavy load-carrying exposure category were characterized by having reported, on average, carrying lighter loads for longer durations/frequencies when compared to those in the “low” exposure category(weight: 69.8 kg (SE: 1.6) versus 57.5 kg (SE: 1.7); frequency: 140.0 loads/week (SE: 343.3) versus 210.0 loads/week (SE: 155.7), respectively).

### 3.2. Symptoms of Lower Abdominal Discomfort Suggestive of POP and POPDI-6 Score

Among the Tanzanian women, almost half reported feelings of heaviness or dullness in the pelvic area, as well as pressure in the lower abdomen (n = 38; 47.5%) and 21 (26.3%) had symptoms consistent with incontinence (Table 3). Twelve women (15.0%) reported that they had a bulge that they could either see or feel in their vaginal area. This was associated with the largest mean increase in the summary score for lower abdominal discomfort (adjusted prevalence difference (PD_a_) = 11.7; 95% CI: 8.4, 15.0).

The most prevalent symptoms of lower abdominal discomfort among women in Nepal were pain while urinating (36.4%), problems holding urine (18.2%), and heaviness/dullness in the pelvic area (18.2%) (Table 3). Among all symptoms, pressure in the lower abdomen (PD_a_ = 36.6; 95% CI: 22.3, 51.0) and heaviness/dullness in the pelvic area (PD_a_ = 31.8 95% CI: 24.9, 38.7) were associated with the largest mean increases in overall discomfort scores.

### 3.3. Load Carrying and Lower Abdominal Discomfort Suggestive of POP

In both study populations, compared to women with lower load-carrying exposure index, those with higher such indexes had larger mean lower abdominal discomfort scores, although confidence intervals included the null in unadjusted and adjusted analyses (Table 4). The magnitude of the association between load carrying and lower abdominal discomfort was higher among study participants in Nepal (PD_a_ = 9.3; 95% CI: −4.9–23.6; *p* = 0.18) compared to Tanzania (PD_a_ = 3.7; 95% CI: −3.8–11.3; *p* = 0.33). Using the best-fitting line for our logistic regression model, with the summary index for load carrying modeled as a continuous variable, a woman with load-carrying exposures corresponding to the average summary index score had 45% and 12.2% predicted probabilities of having high discomfort, in Tanzania and Nepal, respectively (Figure 3 and Figure 4).

## 4. Discussion

In two exploratory cross-sectional studies conducted among in Tanzania and Nepal, we quantified the burden of load carrying experienced by women as part of their daily household and income-generating activities and identified a substantial burden of symptoms of lower abdominal discomfort, with trends suggestive of discomfort increasing with heavy load carrying. Even among women who do not carry loads as a source of income, cultural expectations often dictate heavy load-carrying activities as pervasive parts of their daily activities. In conducting this comparative study, we are contributing to the scant epidemiologic research examining associations between heavy load-carrying activities and lower abdominal discomfort, and, at the same time, drawing attention to the cumulating evidence that heavy load carrying activities are likely important causal factors for the high prevalence of POP among women in developing countries.

The positive associations between increasing load-carrying exposures and lower abdominal discomfort we identified corroborate qualitative reports of heavy load carrying as a perceived cause of POP [5,7,26], and evidence from Ethiopia and Tanzania for significant associations between frequent and sustained load-carrying exposures and prevalence of examination-confirmed anatomical POP [18,27]. Our findings are also consistent with evidence from developed countries, where occupations involving heavy lifting have been shown to have increased odds of genital prolapse compared to the general population [10] and other occupations with less frequent load carrying [11]. Despite the positive trends identified between load carrying and lower abdominal discomfort suggestive of POP in our study, the effect estimates had wide, imprecise confidence intervals that included the null. This may indicate that there is no association between heavy load-carrying activities and POP or that the sample sizes were insufficient. A recent community-based case control study in Nepal, for example, did not identify an association between load-carrying duration (at least 12 h in the last week) or frequency (almost every day, sometimes, not at all) and uterine prolapse confirmed by pelvic examination [19]. However, that study would have involved generally lighter loads than were carried by the sand mining women in the present study. Load-carrying exposures in our analysis incorporated load weight and frequency of load carrying (number of loads carried per week), which may be important determinants of a woman’s likelihood of prolapse.

While our study employed questions from scales validated for the clinical evaluation of POP symptoms, the low specificity of each individual symptom for POP may have resulted in the misclassification of symptoms of menstruation or urinary tract infections (UTIs) as being symptoms of lower abdominal discomfort suggestive of POP. For example, despite high prevalence of examination-confirmed POP among women in Ethiopia, the questionnaire used there to assess symptoms of prolapse showed very low sensitivity, even among those with obvious and visible POP [27]. Our reliance on self-report may have resulted in under-reporting of discomfort symptoms because of social desirability bias, especially when considered in the context of the stigma associated with POP in the largely patriarchal societies from which our two study populations were sampled. Feelings of shame and fear of discrimination and violence have been shown or suspected to impede women’s willingness to seek care or report symptoms of POP in a number of countries, including Nepal [5,6], Ethiopia [27,28], Tanzania [18], and Gambia [29]. If under-reporting of reproductive discomfort occurred non-differentially among groups defined by load-carrying exposures, the association between load carrying and reproductive discomfort is likely to be underestimated.

Measurement of lower abdominal discomfort in the present study was also limited to current symptoms of POP, with no assessment of previous diagnoses or treatment of POP. Inclusion of women who previously received treatment for POP in our study would likely have biased our estimates towards the null. Differing cultural contexts can affect perception and reporting of pain and discomfort, thus posing additional challenges in comparing the findings from these two populations [30]. Future research examining the potential impacts of heavy load carrying on lower abdominal discomfort should preferably utilize longitudinal designs and also incorporate physical examinations to increase the specificity of outcome diagnosis.

The cross-sectional design of our study precluded our ability to examine the likely complex ways in which other important predictors of POP and reproductive discomfort, including factors related to childbirth such as early childbearing, inadequate birthing practices, and the length of time which women are allowed to rest from load-carrying activities after delivery, may interact with heavy load-carrying exposures to impact likelihood of POP. For example, in rural Tanzania, where the average fertility rate is 6 children per woman, only 55% of live births were assisted by a skilled attendant [31]. In Nepal, in 2011, only 36% of live births were assisted by a skilled attendant and women were often expected to resume carrying loads of wood, water or agricultural crops within a few days of giving birth [6,32]. A systematic review of detrimental impacts of load carrying in sub-Saharan Africa recommended a minimum resting recovery time following childbirth until participating in physically demanding activities [14]. It is possible that the potential negative health impacts of heavy load carrying may be heightened in women who have recently given birth. Future research should therefore not only be longitudinal, but have larger sample sizes to explore the ways in which temporal relationships between load carrying, age, and pregnancy may impact associations with reproductive health.

## 5. Conclusions

Findings from our studies underscore the need for further research examining associations between heavy load-carrying practices and reproductive health. Indeed, a stronger evidence base is needed for the implementation of culturally appropriate interventions or preventive measures to mitigate adverse health impacts, such as the ‘silent epidemic’ of POP, among women in LMICs. Such research has the potential to greatly improve the mental and physical health and well-being of those populations.

## Figures and Tables

**Figure 1 ijerph-18-01279-f001:**
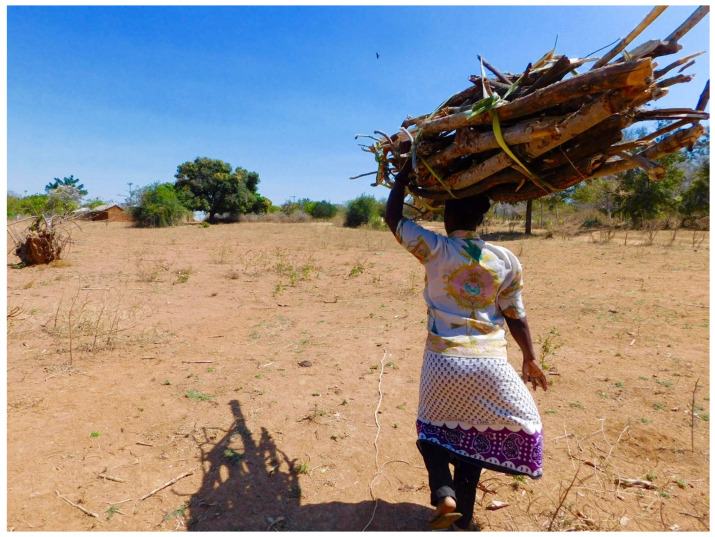
A female carrying a load of wood in Shinyanga region, Tanzania, July–August 2016.

**Figure 2 ijerph-18-01279-f002:**
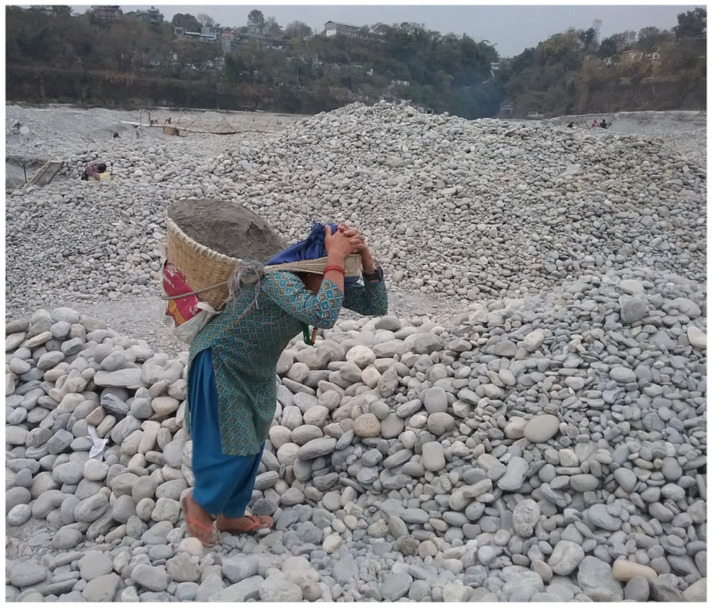
A female sand-miner carrying a load of sand using a doko and namlo at a sand mining site at the Seti River in Pokhara, Kaski district, Nepal, December 2016–January 2017.

**Figure 3 ijerph-18-01279-f003:**
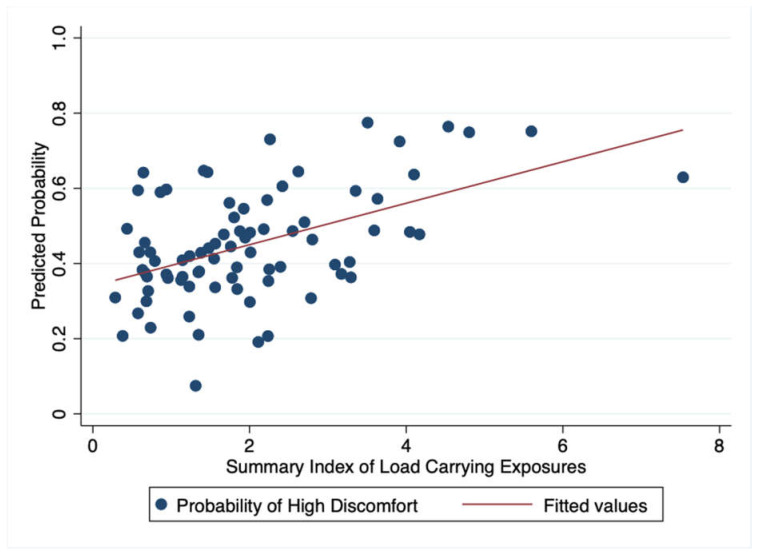
Predicted probability of high discomfort with load-carrying exposures (continuous), Tanzania, July–August 2016.

**Figure 4 ijerph-18-01279-f004:**
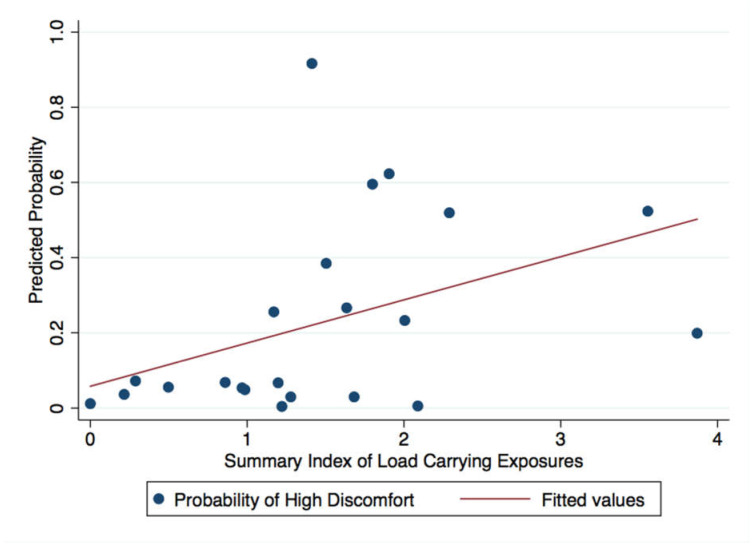
Predicted probability of lower abdominal discomfort with load-carrying exposures (continuous), Nepal, December–January 2017.

**Table 1 ijerph-18-01279-t001:** Characteristics of the study samples of women, Shinyanga Region, Tanzania and Kaski District, Nepal, 2016–2017.

Participant Characteristic	Tanzania (N = 80)		Nepal (N = 22)	
	Mean (SD)	N (%)	Mean (SD)	N (%)
Age (years)	31.6 (12.2)		36.1 (8.1)	
18–30		47 (57.3)		8 (36.4)
>30		26 (31.7)		14 (63.6)
Unknown		9 (11.0)		0 (0.0)
BMI (kg/m^2^)	24.4 (4.2)		24.7 (3.7)	
Underweight (<18.5)		2 (2.4)		1 (4.6)
Normal (18.5–25)		55 (67.1)		11 (50.0)
Overweight (>25)		27 (32.9)		10 (45.5)
Marital status				
Married/with partner		61 (74.4)		17 (77.3)
Unmarried/Single/Divorced/Separated		21 (25.6)		4 (22.7)
Caste/ethnicity ^A^				
Dalit				5 (22.7)
Brahmin/chhetri				1 (4.6)
Janajati				16 (72.7)
Primary Occupation				
Farmer		78 (95.1)		0 (0.0)
Sand miner		0 (0.0)		22 (100.0)
Other		4 (4.9)		0 (0.0)
Years working as a sand miner	-	-		8.4 (9.3)
Parity	4.0 (2.5)		3.4 (2.2)	
0–3 children		45 (54.9)		13 (59.1)
≥4 children		37 (45.1)		9 (40.9)
Number of lifetime pregnancies	4.4 (3.1)		3.9 (2.5)	
None		3 (3.7)		2 (9.1)
1–2 pregnancies		13 (15.9)		6 (27.3)
3–4 pregnancies		33 (40.2)		5 (22.7)
≥5 pregnancies		33 (40.2)		9 (40.9)
Age at first child (years)	18.1 (2.2)		19.7 (4.5)	
13–19 years		42 (52.5)		10 (45.5)
≥19 years		24 (30.0)		10 (45.5)
Missing		16 (20.0)		2 (9.1)

BMI = body mass index; ^A^ Castes originate from the Hindu religion and are hierarchical groups based on social stratification. In Nepali culture, the Dalit represent a low caste group, Brahmin/Chhetri are high caste Hindus, and Janjati are indigenous populations.

**Table 2 ijerph-18-01279-t002:** Summary of load-carrying exposures, Shinyanga Region, Tanzania and Kaski Region, Nepal, 2016–2017.

Characteristic	Tanzania			Nepal		
Median (IQR)	Overall (N = 80)	Low Load Carrying ^A^ (n = 40)	High Load Carrying ^A^ (n = 40)	Overall (N = 22)	Low Load Carrying (n = 11)	High Load Carrying (n = 11)
Weight of load (kgs)	20.0 (13.2–22.2)	19.2 (11.8–21.5)	20.0 (17.3–23.0)	65.5 (57.5–70.1)	69.8 (64.9–79.7)	57.5 (54.9–66.5)
Duration per load (mins) ^B^	30 (15–30)	20 (15–30)	30 (20–35)	0.1 (0.0–0.0)	0.1 (0.0–0.0)	0.1 (0.0–4.0)
Number of loads per week	16.5 (9–32)	11 (5.5–14)	33.5 (22.5–36)	164.5 (120.0–280.0)	140.0 (100.0–175.0)	210.0 (140.0–700.0)

IQR: Interquartile range; ^A^ Summary index of load carrying generated using polychoric principal component analysis of load-carrying duration, frequency, and weight. Dichotomized using the median index score to indicate “low” or “high” load-carrying exposure. ^B^ Self-reported load-carrying duration 0.1 min per load was representative of durations <1 min.

**Table 3 ijerph-18-01279-t003:** Association between symptoms of lower abdominal discomfort suggestive of POP and summary discomfort score, Tanzania and Nepal, 2016–2017.

Symptoms of Lower Abdominal Discomfort	Tanzania (N = 80)			Nepal (N = 22)		
	N (col %)	Unadjusted PD (95% CI)	Adjusted PD (95% CI)	N (col %)	Unadjusted PD (95% CI)	Adjusted ^1^ PD (95% CI)
Pain urinating						
No	32 (39.0)	0.00 (ref)	0.00 (ref)	14 (63.6)	0.00 (ref)	0.00 (ref)
Yes	50 (61.0)	11.9 ** (4.7, 19.1)	11.2 ** (4.1, 18.4)	8 (36.4)	14.9 * (3.7, 26.1)	15.1 * (3.6, 26.7)
Problems holding urine						
No	59 (73.8)	0.00 (ref)	0.00 (ref)	18 (81.8)	0.00 (ref)	0.00 (ref)
Yes	21 (26.3)	9.4 ** (6.7, 12.1)	9.5 ** (6.8, 12.1)	4 (18.2)	1.4 (−15.4, 17.8)	3.4 (−14.2, 20.9)
Pressure in lower abdomen						
No	42 (52.5)	0.00 (ref)	0.00 (ref)	20 (90.9)	0.00 (ref)	0.00 (ref)
Yes	38 (47.5)	11.0 ** (8.4, 13.6)	10.7 ** (8.1, 13.4)	2 (9.1)	33.3 ** (17.7, 49.0)	36.6 ** (22.3, 51.0)
Heaviness/dullness in pelvic area						
No	42 (52.5)	0.00 (ref)	0.00 (ref)	18 (81.8)	0.00 (ref)	0.00 (ref)
Yes	38 (47.5)	11.7 ** (9.1, 14.2)	11.6 ** (8.9, 14.3)	4 (18.2)	31.9 ** (25.0, 38.9)	31.8 ** (24.9, 38.7)
Bulge in vagina						
No	68 (85.0)	0.00 (ref)	0.00 (ref)	21 (95.5)	0.00 (ref)	0.00 (ref)
Yes	12 (15.0)	12.1 ** (8.8, 15.4)	11.7 ** (8.4, 15.0)	1 (4.6)	18.7 (−10.5, 47.8)	12.8 (−23.9, 49.5)

^1^ Adjusted for BMI, age, parity; PD: Prevalence difference; CI: Confidence interval; *p*-value: * *p* < 0.05, ** *p* < 0.01.

**Table 4 ijerph-18-01279-t004:** Association between heavy load carrying and summary lower abdominal discomfort score, Tanzania and Nepal, 2016–2017.

Country	Heavy Load Carrying	Mean Discomfort Score (SD)	Unadjusted PD (95% CI)	Adjusted PD ^1^ (95% CI)
Tanzania	Low	16.4 (14.8)	0.00 (Ref)	0.00 (Ref)
	High	19.8 (18.7)	3.4 (−4.1, 11.0)	3.7 (−3.8, 11.3)
Nepal	Low	4.5 (8.4)	0.00 (Ref)	0.00 (Ref)
	High	13.1 (17.1)	8.50 (−3.5, 20.5)	9.3 (−4.9, 23.6)

^1^ Adjusted for BMI, age, parity; PD: Prevalence difference; CI: Confidence interval.

## Data Availability

The data presented in this study are available on request from the corresponding author. The data are not publicly available to protect participant confidentiality.

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
