# Peer review of "Heavy Load Carrying and Symptoms of Pelvic Organ Prolapse among Women in Tanzania and Nepal: An Exploratory Study"

_ijerph, 2021, doi:10.3390/ijerph18031279_

Round 1

Reviewer 1 Report

Dear authors, accept my sincere thanks for adjusting your manuscript. I hope my previous remarks were beneficial for you. Beside minor suggestions (I propose you change the phrase "activities and reproductive health" in line 266 to "activities and lower abdominal discomfort" and "heavy load carrying on reproductive health" in line 306 to "heavy load carrying on lower abdominal discomfort", as you wisely used in the beginning), I believe that your study is a hopeful prelude for a new, more detailed and prospective one, dealing with all the aspects that I have mentioned, in my humble opinion, in my original remarks. I can only imagine the difficulties you have faced during your study, let alone to design a prospective one, but this is what science is all about, overcoming all obstacles!!!

Keep up your good work and I hope you will produce new studies of great value for the scientific community.

Author Response

Thank you! We have revised line 266 (updated line 266) and line 306 (updated lines 307-308) as suggested.

Reviewer 2 Report

The study by Aybüke Koyuncu et al. etitled ,, Heavy load carrying and symptoms of pelvic organ prolapse among women in Tanzania and Nepal: an exploratory study”

The authors responded to my doubts. They changed the title and also re-edited the text. I also  do agree with the second reviewers’ comments .The ,, Reproductive Health among… -this phrase was probably  an unfortunate shortcut.

In general, the small sample size is the  main but no only the limitation of that study.. Thus  the conclusions must be very carefully formed . However, due to the raising of an important social problem, the work may be accepted in a changed form. I suggest to extend the experiment on wider population

Author Response

Thank you! We hope to leverage our exploratory study to justify broader investment from public health researchers and funders and pursue larger prospective studies in the future.

This manuscript is a resubmission of an earlier submission. The following is a list of the peer review reports and author responses from that submission.

Round 1

Reviewer 1 Report

The current study is an observational cross-sectional one, which compared two groups of women from two developing countries, to examine the association between the women's lifting characteristics and the risk of developing pelvic organ prolapse. The article is excellently designed, with a literature review that presents the problem well, methods that are well planned and described in detail and very clearly. The whole discussion is relevant and addresses all my doubts. I do not see any need for corrections or additions. Well done!

Reviewer 2 Report

Dear authors,

I regret to inform you that, in my humble opinion, your researh have too many flaws in design and execution, beginning with the title which misleads any reader. To be specific, you are referred to reproductive health, when you do not assess it at all.

I have to admit that I am a little bit confused about your intentions. In introduction you mention that your goal is to "increase understanding of the complex relationship between characteristics pf lifting and POP" so I don't find it in order with your title, hence I find it misleading.

If your intention was to identify parameters which contribute to POP and/or poor reproductive health, or prognostic factors of POP and poor reproductive health in these two populations, then I find it intriguing. Nevertheless your research lacks objective tools in assessing reproductive/sexual health, such as the Pelvic Organ Prolapse/Urinary incontinence sexual Questionnaire (PISQ-12) the Female Sexual Function Index (FSFI) etc.

You keep referring to symptoms of reproductive health discomfort, but you neglect to address any issues such as dyspareunia, pain during intercourse, desire in having intercourse, how many participants were pregnant during your study and how many of them had a miscarriage, which surely reflect the status of reproductive health of women in study.

Also, as you admit repeatedly, your study has serious limitations and lacks adequate number of participants, without taking in account my previous remarks and flaws of methods and tools used, in order to extract safe results.

Therefore I suggest you redesign your study. I know that this breaks your feelings, but I repeat, in my humble opinion, you should reconsider drastically your, interesting without any hesitation, yet flawful study.

Reviewer 3 Report

The study by     Aybüke Koyuncu et al. etitled ,,               Heavy Load Carrying and Adverse Reproductive Health among Women in Tanzania and Nepal:”

The study examines the impact of manual labor performed by women in 2 countries on pelvic organ prolapse (POP). It is a very important clinical problem. Contrary to Western European countries, where it occurs frequently after menopause, in developing countries it affects women in younger age groups and is related to the work performed.T he study is interesting, congratulations for figures. ,,Symptoms of reproductive health discomfort were measured using questions from the Pelvic Organ Prolapse Distress Inventory (POPDI-6) validated scale”

However, there are several issues that need to be clarified before publication

  1. Please tell me why you compared these two countries. Probably this problem occurs in another one. Do the origin of authors had an impact?
  2. The introduction although interesting is too long. I think that you can omit without losing sense paragraphs 64-72 and put that to discussion section
  3. Methods section: Again -paragraphs  97-110 should be removed and put into discussion section. Please focus only on  methodology .
  4. How many of women gave the informed consent by signature and how many by thumberprint. Please specify.
  5. It is not clear if ,,wristwatch-like device enabled with GPS capabilities was utilized to assess distance traveled by women over a 24-hour wearing period” was also applied by women in Nepal?
  6. How many of women had cesarean section?  Did some of them had chronic caugh which can lead to  prolapse?
  7. The number of recruited women especially in Nepal is low, that makes the conclusions should be taken with caution. Please comments .Why you did not recruit more women.
  8. You should not begin of discussion with presenting limitations. It is obvious that the weakness of that study is a small sample size to spread out  the conclusions on the entire population could not be appropriate . The reviewer and readers know that  . If you often state that sample size is small I would feel that manuscript is not  enough powered  and you should increase the number of women in the experiment.  In that section you state :

Line 29: ,, Our two populations are small and non-representative subsamples of a wide array of circumstances involving women in heavy physical labor in LMICs”

Line 31: In conducting this small comparative study

Line 42 :This may indicate that there is no association between heavy load carrying activities and reproductive health or that the sample sizes were insufficient

Line 49: Additionally, due to our limited sample size we were unable to explore the

Line 73: Our limited sample sizes and cross-sectional designs

I think that you should change the discussion if that study can be further assessed

 Focus on your results , compare them to another studies ( !do not say that  you have got limited number of patients more than one time). Remove  some informations from the introduction/methods to discussion. Stress that despite limited sample size it is very important social problem. Finally  you conclude t ,, Future research should therefore not only be longitudinal, but have larger sample sizes and should also consider the ways in which temporal relationships between load carrying, age, and pregnancy may impact associations with reproductive health” . It is a good conclusion .